# Unconstrained Models as Constrained Problem Solvers: Duality-Driven Adaptation Without Retraining

## Abstract

We present a novel extension of the forward-backward (FB) representation framework that enables zero-shot constrained reinforcement learning (RL) by embedding both reward and cost functions into a shared latent space. While existing FB methods excel in generalizing across rewards, they fail to account for constraints, a critical limitation in real-world applications where agents must satisfy varying cost budgets or safety requirements. Our approach overcomes this gap through a latent-space reparameterization grounded in Lagrangian duality, allowing efficient inference of constraint-aware policies without requiring any retraining at deployment. By leveraging a latent-space reparameterization grounded in Lagrangian duality, our method allows for efficient inference of constraint-aware policies. Extensive experiments on the ExORL benchmark demonstrate that our method achieves superior task performance while adhering to cost constraints, consistently outperforming prior FB-based and primal-dual baselines. These results highlight the effectiveness and practicality of latent-space constrained policy inference for scalable and safe RL.

## 1 Introduction

Reinforcement learning (RL) has achieved remarkable success in domains such as games, robotics, and decision-making under uncertainty, but its adaptability to changing task specifications remains limited. When the reward function is modified, the policy often requires full retraining, even if the environment transition probabilities remain unchanged, which severely restricts scalability in dynamic real-world settings. To address this challenge, the paradigm of *Zero-Shot RL* (Dayan, 1993; Barreto et al., 2017; 2018) has been developed, aiming to train agents that can immediately adapt to new tasks defined by unseen reward functions without further environment interaction. A particularly promising approach is the *forward-backward (FB) representation framework* (Touati & Ollivier, 2021; Touati et al., 2023; Pirotta et al., 2023; Jeen et al., 2024), where the agent learns a reward-agnostic representation of the environment during pretraining and, at test time, computes a latent vector for a given reward function to directly recover the corresponding optimal policy. This framework enables efficient and scalable adaptation to unseen tasks.

On the other hand, many real-world RL applications require agents to optimize performance while satisfying safety, budget, and risk constraints, making constraint handling a critical requirement for deployment (Altman, 1999; Achiam et al., 2017). Violations can be catastrophic in safety-critical settings such as robotics or autonomous driving (Brunke et al., 2022; Shi et al., 2023), and in resource-limited domains exceeding a budget can invalidate a solution regardless of reward (Bhatia et al., 2019; Wang et al., 2023). In practice, constraints are often formalized as cost functions with certain budgets, and the objective is to maximize expected reward while keeping the cumulative costs below the thresholds (Ghosh et al., 2022; 2024).

While FB generalizes well across rewards, enabling Zero-Shot adaptation, extensions to handle constraints are nontrivial. A seemingly natural FB-based surrogate, proposed by Touati & Ollivier (2021), subtracts cost embeddings from reward embeddings, thereby treating costs as negative rewards. However, this provides no guarantee to budget satisfaction since it does not prevent trading rare but high-cost violations for a higher expected reward. Moreover, a single weighted sum can not

adapt across tasks with different budgets, where different optimal trade-offs are required. As a result, it conflates feasibility with preference and fails to deliver the principled guarantees required for constrained RL.

A second line adopts *primal-dual optimization* for constrained RL. The idea is to build a Lagrangian that penalizes constraint violations via a Lagrange multiplier $\lambda$. The algorithm then alternates updates of the *primal variable* (the policy) and the *dual variable* $\lambda$, maximizing the Lagrangian with respect to the policy and adjusting $\lambda$ to balance reward and cost satisfaction (Ding et al., 2021; Paternain et al., 2022). While a (near-)zero duality gap holds under mild conditions (Paternain et al., 2019), two practical issues occur: (i) the instability and conservativeness that arise during primal-dual training, and (ii) the use of discounted values that obscure whether the budgets are truly met in expectation.

In this paper, we argue that neither explicitly retraining an FB model with primal-dual machinery or folding costs into negative rewards is necessary to enforce budgets. The key insight is that the FB framework already encodes optimal policies across a broad spectrum of reward functions during pretraining. Thus, constrained RL can be reframed more simply: rather than learning both the policy and the multiplier, one only needs to identify the optimal $\lambda$ that enforces the budget. In effect, FB reduces the constrained problem to *tuning* $\lambda$ while reusing the pretrained policy family, enabling constraint enforcement that is both more efficient and more stable than standard primal-dual approaches.

We also highlight a practical challenge: the estimated cost in the model can differ from the actual cost due to the presence or absence of discount factors and various approximation errors. These discrepancies can lead to significant performance issues, especially when constraint thresholds are tight or the model's latent representations are not highly accurate. To address this, we introduce a surrogate threshold, an adjusted version of the original cost limit, estimated from online samples, that is more reliable in the model's latent space. By jointly tuning this surrogate threshold and the Lagrange multiplier through a few trial runs in the environment, we can effectively select a policy that satisfies the actual constraint while maintaining strong performance.

In Section 6, we evaluate our approach using the ExORL benchmark suite (Yarats et al., 2022), which is built on the DeepMind Control Suite (Tassa et al., 2018) and includes a diverse set of tasks spanning locomotion and goal-reaching domains. To measure constraint satisfaction, we introduce a velocity-based safety cost and systematically vary the constraint thresholds across experiments. Our results demonstrate that the proposed method delivers strong task performance while adhering to the specified cost budgets. Importantly, these outcomes are achieved using an FB model that was unaware of costs during pretraining. These findings highlight the effectiveness of our approach as a practical and scalable solution for constrained policy adaptation, showing that high-quality, constraint-aware policies can be recovered without the need for retraining.

## 2 RELATED WORK

### 2.1 CONSTRAINED REINFORCEMENT LEARNING

Constrained RL aims to train agents that maximize expected return while satisfying constraints on cumulative cost or risk signals. A classic formulation is the constrained Markov Decision Process (MDP) (Altman, 1999), which augments the RL objective with constraint functions and associated cost budgets. Numerous works have proposed approaches based on Lagrangian duality (Achiam et al., 2017; Tessler et al., 2018; Chow et al., 2019; Yang et al., 2020; Ding et al., 2021; Paternain et al., 2022), wherein a dual variable (Lagrange multiplier) is updated alongside the policy to enforce constraints. Lagrangian duality methods are theoretically well-founded: recent results establish that constrained RL problems exhibit *zero duality gap* under mild assumptions, ensuring that primal-dual methods can recover optimal constrained solutions (Paternain et al., 2019). However, despite these guarantees, such approaches face notable practical limitations.

One major challenge is *sample inefficiency*. Even recent advances using improved optimization schemes (Mondal & Aggarwal, 2024; Gu et al., 2024a) show that constrained RL typically requires 7-10 times more samples than unconstrained RL to reach comparable performance. This is partly due to the difficulty of balancing reward and cost gradients, which may conflict in practice and hinder stable policy learning (Xu et al., 2021; Gu et al., 2024a). To address this issue, Gu et al.

(2024a) proposed a dynamic sample manipulation strategy based on reward-cost gradient conflict to improve efficiency. However, even with adaptive sampling, constrained RL remains substantially more expensive.

Another critical limitation is the *lack of generalization*. Most constrained RL algorithms are trained for a fixed cost function and budget, and cannot generalize across different constraint specifications without retraining (Stooke et al., 2020; Xu et al., 2022; Brunke et al., 2022; Gu et al., 2024b). This inflexibility restricts their usability in real-world settings where agents must operate under evolving or task-dependent safety profiles.

Our work addresses these limitations by introducing a latent-space formulation that enables post-hoc adaptation to novel cost functions and budget thresholds. Specifically, we build upon the forward-backward representation framework from the Zero-Shot RL literature (Touati & Ollivier, 2021; Pirotta et al., 2023; Touati et al., 2023; Jeen et al., 2024), which is designed to generalize across unseen reward functions without further environment interaction. Leveraging the same reward-agnostic representations that make Zero-Shot adaptation possible, we extend the framework to incorporate constraints. This extension preserves the compositional and efficient adaptation properties of FB, allowing constrained policy optimization to be performed from a single model pretrained without explicitly considering constraints.

## 2.2 ZERO-SHOT REINFORCEMENT LEARNING

Zero-Shot RL aims to train agents that can adapt to new tasks without requiring further environment interaction. Classical approaches in Zero-Shot RL often assume that the reward for each task can be expressed as a linear combination of predefined feature functions (Barreto et al., 2018; 2017; Dayan, 1993), limiting their applicability to tasks with specific reward representations. These methods depend heavily on manually specified features, which are infeasible to scale to complex or continuous task spaces. On the other hand, although goal-conditioned value functions (Schaul et al., 2015; Andrychowicz et al., 2017) allow greater flexibility, they still require explicit specification of goal families during training and cannot adapt to novel reward compositions after training is completed.

In contrast, the FB representation (Touati & Ollivier, 2021; Pirotta et al., 2023; Touati et al., 2023; Jeen et al., 2024) learns a pair of representations of the environment dynamics in a reward-free phase. These representations, denoted as $F$ and $B$, summarize the successor dynamics of all possible policies in latent space. This enables efficient generalization across arbitrary reward functions, including those defined after training, and supports expressive policy composition beyond goal-conditioned methods. To address the inherent challenges of reward-free learning, Sun et al. (2025) proposed an exploration strategy that improves training stability and overall performance. Additionally, Tirinzoni et al. (2025) incorporated demonstrations to guide training and demonstrated that FB-derived embeddings can generalize across a wide range of complex physical tasks, achieving Zero-Shot whole-body humanoid control with a single model.

Despite their success in learning task-generalizable representations, none of these works incorporates generalization across varying cost budgets into the FB framework. That is, FB-based models support adaptation to arbitrary reward functions, they overlook the constrained RL setting where policies must also satisfy diverse budgetary requirements. Our work fills this gap by extending the FB framework to constrained RL via an efficient latent-space primal-dual formulation.

## 3 PRELIMINARY

### CONSTRAINED REINFORCEMENT LEARNING

Constrained RL seeks to learn policies that not only maximize cumulative reward but also satisfy constraints on expected cumulative cost. Formally, given a constrained MDP defined by $(\mathcal{S}, \mathcal{A}, P, r, c, \gamma, \rho_0)$, where $r : \mathcal{S} \times \mathcal{A} \to \mathbb{R}$ is a reward function and $c : \mathcal{S} \times \mathcal{A} \to \mathbb{R}_+$ is a cost function, the Constrained RL objective is (Liu et al., 2021; Malik et al., 2021):

$$\max_{\pi} \quad \mathbb{E}_{s \sim \rho_0} \left[ V^{\pi}(s) \right], \quad \text{subject to} \quad \mathbb{E}_{s \sim \rho_0} \left[ V_c^{\pi}(s) \right] \leq \eta, \tag{1}$$

where $V^{\pi}(s) := \mathbb{E}_{\pi} \left[ \sum_{t=0}^{\infty} \gamma^t r(s_t, a_t) \mid s_0 = s \right]$, and $V_c^{\pi}(s)$ is defined analogously for the cost. The constraint threshold $\eta \in \mathbb{R}_+$ typically encodes safety or resource limits.

This formulation is commonly addressed via Lagrangian duality, a standard technique in constrained optimization (Liu et al., 2020; Chow et al., 2018). Introducing a Lagrange multiplier $\lambda \geq 0$, the associated Lagrangian is given by:

$$\mathcal{L}(\pi, \lambda) = \mathbb{E}_{s \sim \rho_0} \left[ V^\pi(s) - \lambda \left( V_c^\pi(s) - \eta \right) \right]. \tag{2}$$

The resulting optimization is expressed as the following saddle-point problem:

$$\min_\pi \max_{\lambda \geq 0} -\mathcal{L}(\pi, \lambda). \tag{3}$$

## FORWARD-BACKWARD REPRESENTATIONS AS POLICY ENCODERS

Let $M = \langle \mathcal{S}, \mathcal{A}, P, \gamma \rangle$ denote a reward-free MDP, where $\mathcal{S}$ and $\mathcal{A}$ are the state and action spaces, $P(ds' \mid s, a)$ defines the transition dynamics, and $\gamma \in [0, 1)$ is the discount factor. Given an initial state-action pair $(s_0, a_0)$, the *successor measure* of a policy $\pi$ is defined as:

$$M^\pi(s_0, a_0, X) := \sum_{t=0}^\infty \gamma^t \Pr((s_t, a_t) \in X \mid s_0, a_0, \pi), \quad \forall X \subseteq \mathcal{S} \times \mathcal{A}. \tag{4}$$

This quantity describes the discounted visitation frequency over the trajectory induced by $\pi$, and serves as the foundational object approximated in the FB framework. The central insight in the FB approach is that the optimal Q-function for any reward $r : \mathcal{S} \times \mathcal{A} \to \mathbb{R}$ can be expressed using a latent representation. Specifically, the FB representation comprises two learnable functions (Touati & Ollivier, 2021; Pirotta et al., 2023; Touati et al., 2023; Jeen et al., 2024; Sun et al., 2025; Tirinzoni et al., 2025):

$$F : \mathcal{S} \times \mathcal{A} \times \mathbb{R}^d \to \mathbb{R}^d, \qquad B : \mathcal{S} \times \mathcal{A} \to \mathbb{R}^d,$$

which approximate the decomposition of the successor measure. In the standard RL setting, the optimal Q-function can be expressed as $Q_r^* = Mr$, where $M$ denotes the successor operator. The FB representation approximates this by factorizing $M \approx FB$, such that: $Q_r^*(s, a) \approx \left( F(s, a, \cdot) B(\cdot)^\top \right) r$. Given this factorization, we define the latent encoding of the reward function $r$ as $z_r := \mathbb{E}_{(s,a) \sim \rho}[r(s, a) B(s, a)]$, so that the FB model yields the following approximation:

$$Q_r^*(s, a) \approx F(s, a, z_r)^\top z_r, \quad \pi_{z_r}(s) = \arg\max_a F(s, a, z_r)^\top z_r. \tag{5}$$

This structure provides a direct mapping from any reward function to a corresponding policy via a latent vector $z_r$, eliminating the need for planning or iterative optimization at test time.

## ZERO-SHOT ADAPTATION FROM REWARD AND COST EMBEDDINGS

Once the forward and backward models have been pretrained, the FB agent performs zero-shot adaptation by constructing the latent task vector: $z_r = \mathbb{E}_{(s,a) \sim \rho}[r(s, a) B(s, a)]$, which can be estimated from a small number of reward-labeled transitions or computed analytically if $r$ is known. The resulting policy, $\pi_{z_r}(s) = \arg\max_a F(s, a, z_r)^\top z_r$, is then deployed directly without further learning or fine-tuning. However, many real-world applications demand that agents operate under explicit constraints, such as staying within a safety budget or limiting energy consumption. To address this, one might analogously define a cost embedding vector: $z_c = \mathbb{E}_{(s,a) \sim \rho}[c(s, a) B(s, a)]$, where $c(s, a)$ is a task-specific cost function. Touati & Ollivier (2021) suggested incorporating constraints by modifying the latent vector to $z = z_r - z_c$, thereby combining reward and cost into a single direction in the latent space. While this approach is simple, it imposes a fixed trade-off between reward and cost. Crucially, it fails to accommodate task-dependent constraints, such as different users or environments specifying distinct cost budgets $\eta$.

## 4 EXTENDING ZERO-SHOT RL TO CONSTRAINED SETTINGS: PROMISE AND PITFALLS

Zero-shot RL offers a promising alternative by enabling agents to generalize to new reward functions without additional training. This raises a natural question: Can zero-shot RL be extended to constrained settings to enable zero-shot adaptation across varying cost budgets? While such an extension appears theoretically plausible, particularly through formulations based on primal-dual optimization, we demonstrate in Section 4.2 that it struggles in practice due to issues in constraint estimation, enforcement fidelity, and optimization stability.

## 4.1 THEORETICAL EXTENSION OF FB TO CONSTRAINED RL

To enable the FB framework to generalize across different cost budgets, we aim to construct a representation that supports direct, zero-shot policy inference in tasks governed by both reward maximization and cost budget constraints. Specifically, we define a forward encoder that takes as input the reward and cost latent vectors as well as the cost budget $\eta$:

$$F : \mathcal{S} \times \mathcal{A} \times \mathbb{R}^d \times \mathbb{R}^d \times \mathbb{R} \to \mathbb{R}^d, \qquad B : \mathcal{S} \times \mathcal{A} \to \mathbb{R}^d,$$

where $F$ is conditioned on task-specific latent vectors and the constraint level $\eta$, while $B$ provides a shared, task-independent encoding of state-action pairs.

Given a reward function $r$ and a cost function $c$, their respective latent embeddings are defined as:

$$z_r := \mathbb{E}_{(s,a)\sim\rho}[r(s,a)B(s,a)], \qquad z_c := \mathbb{E}_{(s,a)\sim\rho}[c(s,a)B(s,a)], \tag{6}$$

where $\rho$ denotes a fixed state-action distribution used during the reward-free pretraining phase. We then approximate the Q-functions and corresponding constrained policies as:

$$Q_r^*(s,a) \approx F(s,a,z_r,z_c,\eta)^\top z_r, \quad \text{subject to} \quad F(s,a,z_r,z_c,\eta)^\top z_c \le \eta, \tag{7}$$

$$\pi_{z_r,z_c,\eta}(s) = \arg\max_a F(s,a,z_r,z_c,\eta)^\top z_r, \quad \text{subject to} \quad F(s,a,z_r,z_c,\eta)^\top z_c \le \eta. \tag{8}$$

This formulation allows zero-shot deployment in unseen constrained tasks by estimating the latent vectors $z_r$ and $z_c$, and computing the policy $\pi_{z_r,z_c,\eta}$ through a single forward pass. This avoids the need for online constraint-sensitive optimization at deployment time. The overall constrained optimization objective becomes:

$$\max_\pi \mathbb{E}_{s\sim\rho_0} \left[ F(s,\pi(s),z_r,z_c,\eta)^\top z_r \right], \quad \text{subject to} \quad \mathbb{E}_{s\sim\rho_0} \left[ F(s,\pi(s),z_r,z_c,\eta)^\top z_c \right] \le \eta. \tag{9}$$

Using Lagrangian relaxation, we convert the problem into a min-max saddle-point formulation:

$$\min_{\lambda\ge0} \max_\pi \mathbb{E}_{s\sim\rho_0} \left[ F(s,\pi(s),z_r,z_c,\eta)^\top z_r - \lambda(z_r,z_c,\eta) \left( F(s,\pi(s),z_r,z_c,\eta)^\top z_c - \eta \right) \right]. \tag{10}$$

The Lagrangian objective enables us to train a zero-shot constrained RL agent using the FB framework, allowing efficient policy inference that satisfies cost constraints without further environment interaction.

**Theorem 4.1** (Constrained RL Representations Encode All Optimal Policies). *Let* $(F, B, \pi_{z_r,z_c,\eta})$ *be a constrained RL representation of a reward-free MDP with respect to* $\rho$. *For any bounded reward function* $r$ *and cost function* $c$, *define:*

$$z_R := \int r(s,a)B(s,a)\rho(ds,da), \qquad z_C := \int c(s,a)B(s,a)\rho(ds,da). \tag{11}$$

*Then the policy* $\pi_{z_R,z_C,\eta}$ *is an optimal constrained policy, and the associated Q-function is given by:*

$$Q^*(s,a) = F(s,a,z_R,z_C,\eta)^\top z_R - \lambda(z_R,z_C,\eta) \cdot F(s,a,z_R,z_C,\eta)^\top z_C.$$

## 4.2 ZERO-SHOT CONSTRAINED RL FAILS IN PRACTICE

The preceding results demonstrate that the FB framework, when extended to embed both rewards and constraints, is theoretically capable of representing the optimal solution to any bounded constrained RL problem. However, although the constrained RL representation described in Section 4.1 is sufficiently expressive to encode optimal policies, this formulation may not lead to reliable zero-shot constrained learning in practice. In particular, it suffers from several key limitations, including instability in optimization, weak constraint enforcement, and approximation error.

**Limitation 1. Instability and Conservativeness of Primal-Dual Training.** Jointly optimizing the policy $\pi$ and the dual variable $\lambda$ using alternating updates introduces optimization instability (Achiam et al., 2017; Chow et al., 2018) and high sample complexity (Mondal & Aggarwal, 2024; Gu et al., 2024a; Le et al., 2019; Stooke et al., 2020). Even when the value estimates are accurate, the updates to $\lambda$ may oscillate or converge slowly. Moreover, such methods tend to produce overly conservative policies that aim to satisfy constraints across all possible latent conditions $z$, even when constraints are rarely activated.

**Limitation 2. Discounted Values Obscure True Constraint Violation.** In practice, value functions typically estimate cumulative discounted costs $C_z := \mathbb{E}_{\pi_z} \left[ \sum_{t=0}^{\infty} \gamma^t c(s_t, a_t) \right]$ (Tessler et al., 2018; Achiam et al., 2017; Altman, 1999). However, satisfying the constraint $C_z \leq \eta$ under discounting does not imply that the true cumulative cost $\sum_{t=0}^{T} c(s_t, a_t)$ respects the same limit (Geibel, 2006). This mismatch renders discounted estimates unreliable for enforcing hard constraints.

**Limitation 3. Function Approximation Error Compromises Safety.** In most practical settings, value functions $V^\pi$ and $V_c^\pi$ are approximated using neural networks. Let $\hat{C}_z$ be the approximated expected cost: $\hat{C}_z = \mathbb{E}_{s \sim \rho_0} [\hat{V}_c^\pi(s)]$ (Zhang et al., 2024; Saglam et al., 2021). Hence $\hat{C}_z \leq \eta$ does not mean $C_z \leq \eta$. This undermines the reliability of zero-shot adaptation under strict safety conditions.

The challenges mentioned above reveal that, while zero-shot constrained RL is theoretically feasible, its practical realization is hindered by issues in cost estimation, constraint fidelity, and optimization instability. This motivates our proposed alternative: an efficient approach that exploits latent structure while allowing adaptive constraint satisfaction through limited interaction.

## 5 DUALITY-DRIVEN ADAPTATION WITHOUT RETRAINING

To overcome the challenges of zero-shot constrained RL, we leverage the expressiveness of the unconstrained FB latent representation to enable efficient and reliable constraint satisfaction. Our key insight is to reduce the complexity of the primal-dual constrained optimization problem by reformulating it as a simpler latent-space search, requiring only minimal online calibration.

### 5.1 LATENT-SPACE SIMPLIFICATION VIA PRIMAL-DUAL REPARAMETERIZATION

In Section 4.1, the formulation addresses the constrained problem by jointly updating the models $F$ and $B$, policy $\pi$, and Lagrange multiplier $\lambda$. However, it suffers from optimization instability due to the simultaneous updates. To address this, we revisit the standard Lagrangian relaxation approach and introduce a novel reformulation. Let us consider the following constrained optimization problem:

$$\max_\pi \mathbb{E} \left[ \sum_{t=0}^{\infty} \gamma^t R(s_t, \pi(s_t)) \right] \quad \text{s.t.} \quad \mathbb{E} \left[ \sum_{t=0}^{\infty} \gamma^t C(s_t, \pi(s_t)) \right] \leq \eta. \tag{12}$$

This problem can be equivalently formulated as a dual optimization problem:

$$\min_{\lambda \geq 0} \left[ \max_\pi \mathbb{E} \left[ \sum_{t=0}^{\infty} \gamma^t \left( R(s_t, \pi(s_t)) - \lambda C(s_t, \pi(s_t)) \right) \right] + \lambda \eta \right], \tag{13}$$

where $\lambda \geq 0$ is the Lagrange multiplier associated with the cost constraint.

While solving the min-max problem in Eq. equation 13 typically requires joint optimization over policies and multipliers, we offer the key insight that enable a more efficient solution.

**Reward Reparameterization.** For any fixed $\lambda$, solving the constrained problem is equivalent to solving an unconstrained RL problem with a modified reward:

$$R_\lambda(s, a) := R(s, a) - \lambda C(s, a). \tag{14}$$

**Key Insight: Leveraging Optimal Policies through Pretrained FB.** According to Theorem 2 in Touati & Ollivier (2021), the FB framework is capable of representing optimal policies for arbitrary reward functions. This implies that, for each value of $\lambda$, we can obtain the corresponding optimal policy $\pi_{z_\lambda}^*$ without retraining. Unlike conventional constrained RL settings, where neither the optimal policy nor the optimal Lagrange multiplier is known in advance and must be jointly optimized, our approach benefits from having access to the entire family of optimal policies indexed by $\lambda$. **As a result, the constrained optimization problem, originally requiring simultaneous optimization over both policy and multiplier, is reduced to a much simpler task: identifying the optimal Lagrange multiplier $\lambda \in \mathbb{R}_+$ that satisfies the cost constraint.**

The insight enables us to leverage the FB model pretrained without considering cost budgets, denoted by $\bar{F}, \bar{B}, \bar{\pi}$, for constraint-aware policy inference. In particular, the latent representation associated

with the modified reward $R_\lambda$ becomes: $z_\lambda = \mathbb{E}_{(s,a)\sim\rho}\left[R_\lambda(s,a)\bar{B}(s,a)\right] = z_r - \lambda z_c$, where $z_r$ and $z_c$ are the latent embeddings of the reward and cost, respectively.

Given this reparameterization, we approximate the action-value functions of the policy $\bar{\pi}_{z_\lambda}$ as:

$$Q_r^{\bar{\pi}_{z_\lambda}}(s,a) \approx \bar{F}(s,a,z_\lambda)^\top z_r, \quad Q_c^{\bar{\pi}_{z_\lambda}}(s,a) \approx \bar{F}(s,a,z_\lambda)^\top z_c. \tag{15}$$

This formulation enables us to reduce constrained policy inference to the task of selecting an appropriate $\lambda \in \mathbb{R}_+$, allowing for efficient and scalable adaptation without retraining.

## 5.2 LATENT-SPACE LAGRANGIAN MULTIPLIER SEARCH

Building on the latent-space formulation in Section 5.1, we now describe how to efficiently search for the appropriate Lagrange multiplier $\lambda$ that ensures constraint satisfaction. As we vary $\lambda$, the expected cost $\bar{F}^\top z_c$ changes. This allows us to apply a *bisection method* to identify the appropriate $\lambda$ such that the induced policy satisfies the constraint threshold $\eta$. Specifically, we iteratively adjust $\lambda$ by observing whether the predicted surrogate cost $\bar{F}^\top z_c$ is above or below the constraint $\eta$, and update $\lambda$ accordingly. This avoids the need for full optimization and provides a simple yet effective mechanism for constraint-aware adaptation (Algorithm 1 in Appendix).

## 5.3 ONLINE CALIBRATION

While the surrogate cost $\bar{F}^\top z_c$ provides an efficient approximation, it may differ from the true cumulative cost incurred during real-world execution (see Limitations 2 and 3 in Section 4.2). To bridge this gap, we introduce a surrogate budget $\eta'$, representing that when $\bar{F}^\top z_c \leq \eta'$, the actual environmental cost $C_{\text{real}}$ satisfies the real constraint $\eta$. The gap between the surrogate cost and the actual cost can be seen in Table 1.

| | $\eta' = 100$ | | $\eta' = 0$ | | $\eta' = -100$ | | $\eta' = -200$ | |
| --- | --- | --- | --- | --- | --- | --- | --- | --- |
| | $C_{\text{real}}$ | $\bar{F}^\top z_c$ | $C_{\text{real}}$ | $\bar{F}^\top z_c$ | $C_{\text{real}}$ | $\bar{F}^\top z_c$ | $C_{\text{real}}$ | $\bar{F}^\top z_c$ |
| stand | 457 | -23 | 457 | -23 | 269 | -99 | 118 | -199 |
| run | 391 | 28 | 296 | -0.3 | 159 | -100 | 69 | -199 |
| walk | 296 | 53 | 253 | -0.4 | 134 | -100 | 11 | -200 |
| Flip | 421 | 16 | 411 | -0.3 | 404 | -99 | 124 | -200 |

Table 1: This table reports the actual environmental cost observed across tasks when the surrogate budget $\eta'$ is used as the constraint threshold. It highlights the discrepancy between the surrogate cost, $\bar{F}^\top z_c$, and the true cumulative cost $C_{\text{real}}$ incurred during evaluation. Notably, when $F^\top z_c \ll \eta'$, the constraint is inherently satisfied, resulting in a Lagrange multiplier $\lambda = 0$; that is, no additional cost penalty is needed to meet the constraint.

We propose an efficient calibration strategy that operates under a total online sampling budget of $T \times n$ episodes. In each iteration $t = 1, \ldots, T$, we first use a bisection search to find a latent vector $z_\lambda$ such that the surrogate cost $\bar{F}^\top z_c \approx \eta'$. We then evaluate the resulting policy $\bar{\pi}_{z_\lambda}$ by deploying it for $n$ episodes and measuring the average realized cost: $\bar{C}_{\text{real}}^{(t)} = \frac{1}{n}\sum_{i=1}^n C_i^{(t)}$. Based on whether $\bar{C}_{\text{real}}^{(t)}$ is greater or smaller than the desired budget $\eta$, we adjust the surrogate threshold $\eta'$ accordingly. This feedback loop continues until the sampling budget is exhausted. After all $T$ iterations, we select the $\lambda$ that yields an empirical cost closest to the target constraint. This procedure is formalized in Algorithm 2, provided in the Appendix.

**Remark 5.1** (Optional Statistical Guarantee via Hoeffding). *In this work, we limit the number of online samples to ensure calibration efficiency. However, if statistical guarantees are desired, one can incorporate Hoeffding's inequality to determine whether the observed cost is sufficiently close to the expected cost with high confidence. Let $C_1, \ldots, C_n$ be the costs observed in $n$ episodes of executing a fixed policy. Assuming each cost is bounded in $[0, C_{\max}]$, the empirical average satisfies:*

$$|\bar{C}_{real} - \mathbb{E}[C_{real}]| \leq \epsilon, \quad where \ \epsilon := C_{\max}\sqrt{\frac{\log(2/\delta)}{2n}},$$

*with probability at least $1 - \delta$. This bound can be used as a confidence-aware stopping rule for selecting $\lambda$, complementing our fixed-budget approach. This procedure is formalized in Algorithm 3, provided in the Appendix.*

### 5.4 IMPLEMENTATION DETAILS

Our approach enables the use of an unconstrained FB model to satisfy constraints without requiring any retraining. Specifically, we utilize the FB model introduced by Jeen et al. (2024). To obtain the pretrained model, we adhere strictly to their original network architecture and hyperparameter settings, without introducing any additional loss functions, auxiliary networks, or hyperparameter tuning beyond what is necessary for FB pretraining.

The bisection-based solver operates entirely in the latent space and is executed post hoc using a small number of cost evaluations in the target environment, ensuring both computational efficiency and reproducibility. For online calibration, we use a sample budget of $T = 30$ iterations with $n = 1$ episode per iteration, and 1000 transitions per episode, resulting in a total of 30K online samples.

## 6 EVALUATIONS

### 6.1 BASELINES AND EXPERIMENT SETUP

To the best of our knowledge, constrained policy adaptation has not been systematically explored within the Zero-Shot RL framework. To evaluate our approach, we compare it against three variants of the FB framework: FB (Touati & Ollivier, 2021), FB($z_r - z_c$), and FB (Primal-dual). The standard FB baseline considers only reward representations. FB($z_r - z_c$) implements the latent vector subtraction method proposed by Touati & Ollivier (2021), where the combined latent representation is defined as $z = z_r - z_c$. CRL_FB refers to our implementation of the primal-dual approach described in Section 4.1. All models were pretrained using the publicly available codebase from Jeen et al. (2024). Note that the models of FB, FB($z_r - z_c$), and our method were identical in the comparison. The three methods differ only in the input representations.

We conducted experiments using the ExORL benchmark datasets (Yarats et al., 2022), which comprise data collected by unsupervised exploratory algorithms on the DeepMind Control Suite (Tassa et al., 2018). Following prior work (Touati et al., 2023; Jeen et al., 2024), we selected three representative domains: Walker, Quadruped, and Maze. These encompass two locomotion tasks and one goal-reaching task. Within each domain, we evaluated performance across all tasks provided in the DeepMind Control Suite, totaling 13 tasks. To pretrain the FB models, we use datasets collected via Random Network Distillation (RND) (Burda et al., 2019), along with an additional 1 million online samples, resulting in a total of 2 million training samples. This setup aligns with the total sample budget used in Sun et al. (2025). To incorporate safety considerations, we adopted the velocity-based safety constraint setting as described in Gu et al. (2024a). All experiments were repeated five times to ensure statistical robustness, and all models are trained on NVIDIA RTX 4090 GPUs (more experiment settings are listed in Appendix A.3).

### 6.2 RESULTS

Table 2 summarizes the performance and cost metrics, averaged over five random seeds, under a budget constraint of 100. Notably, FB($z_r - z_c$) does not consider the budget during policy inference, limiting its ability to adapt to varying budget levels. CRL_FB, while capable of zero-shot generalization to different constraints, tends to be overly conservative due to the need to satisfy all potential constraint scenarios, leading to suboptimal performance. In contrast, by leveraging a targeted search over the Lagrange multiplier $\lambda$, our method can effectively utilize the available budget and achieve superior task performance.

While both CRL_FB and our method are constraint-aware, our approach offers significant advantages. Specifically, it leverages the standard FB model pretrained without considering constraints, thereby avoiding the stability issues associated with joint optimization. Although both FB and CRL_FB are trained using 2 million samples, CRL_FB requires simultaneous optimization over reward and cost

representations, which leads to substantially longer training times (18 hours vs. 40 hours) and often results in overly conservative behaviors that hinder task performance.

Our method includes a lightweight online calibration step, which uses an additional 30K samples for adjusting the surrogate budget $\eta'$. Since this step does not modify the parameters of the forward or backward models ($F$ and $B$), it can be completed within a few minutes. Importantly, since surrogate cost estimates can deviate from actual costs due to discounting factors and approximation errors, all baselines require such calibration to reliably satisfy real-world constraints.

| Environment | FB | | FB (zr - zc) | | CRL_FB | | Ours | |
|---|---|---|---|---|---|---|---|---|
| Walker | Performance | Cost | Performance | Cost | Performance | Cost | Performance | Cost |
| stand | 799±33 | 427±227 | 433±169 | 68±36 | 125±20 | 0±0 | 672±196 | 82±10 |
| walk | 531±117 | 327±143 | 90±57 | 45±79 | 21±4 | 0±0 | 204±154 | 70±24 |
| run | 215±41 | 412±136 | 50±28 | 28±28 | 20±2 | 0±0 | 136±64 | 94±4 |
| Flip | 403±46 | 360±40 | 135±155 | 83±151 | 21±1 | 0±0 | 317±105 | 75±23 |
| Quadruped | | | | | | | | |
| Stand | 795±122 | 366±236 | 187±187 | 15±18 | 246±187 | 0.44±0.93 | 737±419 | 89±7 |
| Roll | 805±85 | 479±305 | 167±113 | 19±17 | 157±113 | 0.44±0.93 | 761±153 | 81±16 |
| Roll Fast | 484±60 | 463±290 | 123±83 | 28±37 | 128±96 | 0.44±0.94 | 427±255 | 65±33 |
| Jump | 628±80 | 404±258 | 161±122 | 69±75 | 181±136 | 0.45±0.93 | 542±329 | 91±8 |
| Escape | 69±12 | 433±262 | 9±14 | 65±112 | 5±4 | 0.43±0.95 | 41±33 | 73±25 |
| Maze | | | | | | | | |
| Top Left | 933±23 | 85±35 | 41±60 | 245±125 | 17±22 | 0±0 | 710±463 | 80±14 |
| Top Right | 129±258 | 122±137 | 0±0 | 242±247 | 0±0 | 0±0 | 125±251 | 66±8 |
| Bottom Left | 20±41 | 106±114 | 1±1 | 732±404 | 0±0 | 0±0 | 0±0 | 62±22 |
| Bottom Right | 0±0 | 92±72 | 0±0 | 99±107 | 0±0 | 0±0 | 0±0 | 86±7 |

Table 2: Performance and cost comparison across 13 ExORL tasks under a cost budget of 100. Our method consistently achieves strong performance while adhering to the constraint, outperforming prior FB-based approaches. Since the budget is fixed at 100, **the objective is not to minimize cost arbitrarily, but rather to utilize it as effectively as possible, ideally approaching the budget without exceeding it.**

### 6.3 ABLATION STUDIES

To further evaluate the effectiveness of our proposed method, we conducted ablation studies under varying cost budgets and statistical guarantee via Hoeffding Bound. These experiments are essential for providing deeper insights into the behavior of our approach, elucidating its strengths, and identifying areas for potential improvement. Through this analysis, we aim to demonstrate the adaptability of our method and validate its applicability across a wide range of safe RL scenarios. Due to space limitations, detailed results and discussion of the ablation studies are presented in Appendix A.4.

## 7 CONCLUSIONS

We have introduced a novel extension of the FB representation framework to the constrained RL setting, enabling zero-shot adaptation to tasks specified by both reward functions and cost constraints. By embedding both rewards and constraints into a shared latent space and formulating policy inference as a latent-space optimization problem, our method allows for efficient adaptation without retraining. This approach bridges the gap between zero-shot RL and constrained RL, offering a unified framework for safe and flexible policy deployment in dynamic environments. Through comprehensive experiments on the ExORL benchmark, we demonstrate that our method consistently satisfies cost budgets while achieving strong task performance, significantly outperforming existing FB-based and primal-dual constrained RL baselines. These results highlight the potential of latent-space representations for scalable and constraint-aware generalization in reinforcement learning.

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

# A    APPENDIX

## A.1    ALGORITHMS

---

**Algorithm 1** Bisection Search for Lagrange Multiplier $\lambda$

---

1: **Input:** reward embedding $z_r$, cost embedding $z_c$, constraint threshold $\eta$, tolerance $\epsilon$
2: Initialize $\lambda_{\min} \leftarrow 0$, $\lambda_{\max} \leftarrow \Lambda$ (a large enough upper bound)
3: **while** $\lambda_{\max} - \lambda_{\min} > \epsilon$ **do**
4:    $\lambda \leftarrow (\lambda_{\min} + \lambda_{\max})/2$
5:    $z_\lambda \leftarrow z_r - \lambda z_c$
6:    Compute surrogate cost estimate: $\hat{C}_\lambda \leftarrow F^\top z_c$
7:    **if** $\hat{C}_\lambda > \eta$ **then**
8:       $\lambda_{\min} \leftarrow \lambda$
9:    **else**
10:       $\lambda_{\max} \leftarrow \lambda$
11:    **end if**
12: **end while**
13: **return** $\lambda$, $z_\lambda$, $\pi_{z_\lambda}$

---

**Algorithm 2** Efficient Constrained RL via Latent Calibration and Online Evaluation

---

1: **Given:** Budget $\eta$, pre-trained FB model with $\mathbf{F}, \mathbf{B}$, total online sampling budget $T \times n$, cost bound $C_{\max}$
2: Initialize $\lambda \leftarrow$ random, $\eta' \leftarrow \eta$
3: Initialize history buffer $\mathcal{H} \leftarrow \emptyset$
4: **for** $t = 1$ to $T$ **do**
5:    **repeat**
6:       Compute $z_\lambda \leftarrow z_r - \lambda z_c$
7:       Evaluate latent cost estimate: $\hat{C}_{\text{latent}} \leftarrow \mathbb{E}_{s \sim \rho_0}[\mathbf{F}^\top z_c]$
8:       **if** $\hat{C}_{\text{latent}} > \eta'$ **then**
9:          Increase $\lambda$
10:       **else if** $\hat{C}_{\text{latent}} < \eta'$ **then**
11:          Decrease $\lambda$
12:       **end if**
13:    **until** $\hat{C}_{\text{latent}} < \eta' - \epsilon_{\text{latent}}$
14:    Deploy $\pi_{z_\lambda}$ for $n$ episodes and collect costs $\{C_i^{(t)}\}_{i=1}^n$
15:    Compute empirical average $\bar{C}_{\text{real}}^{(t)} = \frac{1}{n} \sum_{i=1}^n C_i^{(t)}$
16:    Store $(\lambda, \bar{C}_{\text{real}}^{(t)})$ in $\mathcal{H}$
17:    **if** $\bar{C}_{\text{real}}^{(t)} > \eta$ **then**
18:       Decrease $\eta'$
19:    **else if** $\bar{C}_{\text{real}}^{(t)} < \eta$ **then**
20:       Increase $\eta'$
21:    **end if**
22: **end for**
23: Select $(\lambda^*, \bar{C}^*) \in \mathcal{H}$ such that $|\bar{C}^* - \eta|$ is minimized and $\bar{C}^* < \eta$
24: **return** Calibrated $z_{\lambda^*}$, policy $\pi_{z_{\lambda^*}}$

---

---

**Algorithm 3** Efficient Constrained RL via Latent Calibration and Hoeffding Bound

---

1: **Given:** Budget $\eta$, pre-trained FB model with $\mathbf{F}$, $\mathbf{B}$, confidence level $\delta$, cost bound $C_{\max}$
2: Initialize $\lambda \leftarrow$ random, $\eta' \leftarrow \eta$, number of episodes $n \leftarrow 0$
3: **repeat**
4:     **repeat**
5:         Compute $z_\lambda \leftarrow z_r - \lambda z_c$
6:         Evaluate latent cost estimate: $\hat{C}_{\text{latent}} \leftarrow \mathbb{E}_{s \sim \rho_0}[\mathbf{F}^\top z_c]$
7:         **if** $\hat{C}_{\text{latent}} > \eta'$ **then**
8:             Increase $\lambda$
9:         **else if** $\hat{C}_{\text{latent}} < \eta'$ **then**
10:         Decrease $\lambda$
11:         **end if**
12:     **until** $\hat{C}_{\text{latent}} < \eta' - \epsilon_{\text{latent}}$
13:     Deploy $\pi_{z_\lambda}$ for $n$ episodes and collect costs $\{C_i\}_{i=1}^n$
14:     Compute empirical average $\bar{C}_{\text{real}} = \frac{1}{n} \sum_{i=1}^n C_i$
15:     Compute Hoeffding bound: $\epsilon \leftarrow C_{\max} \sqrt{\frac{\log(2/\delta)}{2n}}$
16:     **if** $\bar{C}_{\text{real}} > \eta + \epsilon$ **then**
17:         Decrease $\eta'$
18:     **else if** $\bar{C}_{\text{real}} < \eta - \epsilon$ **then**
19:         Increase $\eta'$
20:     **end if**
21: **until** $\bar{C}_{\text{real}} \leq \eta - \epsilon$
22: **return** Calibrated $z_\lambda$, policy $\pi_{z_\lambda}$

---

### A.2 PROOF OF THEOREM 4.1

*Proof.* Following the proof structure for the unconstrained FB representation in Touati & Ollivier (2021), we use an extended model of the successor state distribution. Specifically, we approximate the goal-conditioned occupancy measure $m^\pi(s, a, g)$ using a bilinear component plus a residual:

$$m^\pi(s, a, g) = F(s, a, z_R, z_C, \eta)^\top B(g) + \bar{m}(s, z_R, z_C, \eta, g),$$

where $F$ and $B$ are learnable encoders, and $\bar{m}$ is an action-independent residual that captures parts of the occupancy measure not modeled by the rank-limited term $F^\top B$. This residual term enables a more expressive approximation while restricting only the advantage component to be low-rank.

Let $r(s, a)$ and $c(s, a)$ be any bounded functions. Define their latent embeddings:

$$z_R := \int r(s, a) B(s, a)\, \rho(ds, da), \qquad z_C := \int c(s, a) B(s, a)\, \rho(ds, da).$$

We consider a linear combination in the latent space using Lagrangian relaxation:

$$z_\lambda := z_R - \lambda(z_R, z_C, \eta)\, z_C, \quad \lambda(z_R, z_C, \eta) \geq 0.$$

For any policy $\pi$, let $m^\pi(s, a, g)$ denote the state-action-goal occupancy density, with $g = \varphi(s, a)$ and $g \sim \rho$. Then the FB representation yields:

$$m^\pi(s, a, g) = F(s, a, z_R, z_C, \eta)^\top B(g) + \bar{m}(s, z_R, z_C, \eta, g).$$

Define the surrogate reward $r_\lambda(g) := r(g) - \lambda(z_R, z_C, \eta)\, c(g)$. Then the Q-function under this surrogate becomes:

$$Q^\pi(s, a) = \int r_\lambda(g)\, m^\pi(s, a, g)\, \rho(dg)$$

$$= \int [r(g) - \lambda(z_R, z_C, \eta)\, c(g)] \left[ F(s, a, z_R, z_C, \eta)^\top B(g) + \bar{m}(s, z_R, z_C, \eta, g) \right] \rho(dg)$$

$$= F(s, a, z_R, z_C, \eta)^\top \int [r(g) - \lambda(z_R, z_C, \eta)\, c(g)] B(g)\, \rho(dg) + \int r_\lambda(g)\, \bar{m}(s, z_R, z_C, \eta, g)\, \rho(dg)$$

$$= F(s, a, z_R, z_C, \eta)^\top z_\lambda + \bar{V}^{z_\lambda}(s),$$

where $\bar{V}^{z_\lambda}(s)$ absorbs the residual term independent of $a$. The corresponding residual value function is defined as:

$$\bar{V}^{z_\lambda}(s) := \int \bar{m}(s, z_R, z_C, \eta, g)\, r_\lambda(g)\, \rho(dg),$$

which vanishes when $\bar{m} = 0$.

Define the policy:

$$\pi_{z_\lambda, \eta}(s) := \arg\max_a F(s, a, z_R, z_C, \eta)^\top z_\lambda.$$

Then, since $\bar{V}^{z_\lambda}(s)$ is constant in $a$, this policy maximizes the Q-function.

By strong duality in constrained MDPs, there exists $\lambda^* = \lambda(z_R, z_C, \eta) \geq 0$ such that the policy $\pi_{z_{\lambda^*}, \eta}$ satisfies the constraint:

$$\mathbb{E}_{s \sim \rho_0} \left[ F(s, \pi_{z_{\lambda^*}, \eta}(s), z_R, z_C, \eta)^\top z_C \right] = \eta.$$

Thus, the policy $\pi_{z_{\lambda^*}, \eta}$ is optimal for the original constrained problem. Hence, the latent embedding and FB representation encode all optimal constrained policies. $\qquad\square$

### A.3 EXPERIMENT SETTINGS

#### A.3.1 EXORL DOMAINS

**Walker.** This environment involves a two-legged robot that must perform locomotion starting from a bent-knee position. The state and action spaces are 24- and 6-dimensional, respectively, encompassing joint torques, velocities, and positions. The ExORL benchmark defines four tasks for this domain: `stand`, `walk`, `run`, and `flip`. The `stand` task incentivizes an upright posture with extended legs, while `walk` and `run` build upon this by additionally rewarding forward motion, with `run` placing greater emphasis on speed. The `flip` task encourages angular motion of the torso after standing. All tasks provide dense reward signals.

**Quadruped.** This environment features a four-legged robot required to perform locomotion inside a 3D maze. The state space is 78-dimensional and the action space is 12-dimensional, consisting of joint torques, velocities, and positions. ExORL provides five tasks: `stand`, `roll`, `roll fast`, `jump`, and `escape`. The `stand` task rewards a minimum torso height and straightened legs. The `roll` and `roll fast` tasks involve flipping from a back position, with the latter emphasizing speed. The `jump` task encourages vertical displacement, and the `escape` task requires the agent to navigate out of the maze. Rewards are dense across all tasks.

**Maze.** This domain consists of a 2D environment divided into four rooms, where the agent must move a point-mass to a designated goal room. The state space is 4-dimensional (positions and velocities in $x$ and $y$), and the action space is 2-dimensional (forces in $x$ and $y$). ExORL defines four goal-reaching tasks: `top left`, `top right`, `bottom left`, and `bottom right`. The mass is initialized in the top-left room, and the agent receives a sparse reward that is only provided when it is sufficiently close to the specified goal.

#### A.3.2 HYPERPARAMETERS

See Table 3.

### A.4 ABLATION STUDY

#### A.4.1 VARIOUS COST BUDGETS

To evaluate whether our method can handle varying cost budgets, we conducted experiments with budget levels set to 150, 100, and 50. As shown in Table 4, our method successfully produces feasible solutions across all budget settings. Moreover, we observe that higher budget thresholds generally lead to improved task performance.

| Hyperparameter | Value |
|---|---|
| Latent dimension $d$ | 50 (100 for maze) |
| $F$ / $\psi$ dimensions | (1024, 1024) |
| $B$ / $\varphi$ dimensions | (256, 256, 256) |
| Preprocessor dimensions | (1024, 1024) |
| Std. deviation for policy smoothing $\sigma$ | 0.2 |
| Truncation level for policy smoothing | 0.3 |
| Learning steps | 1,000,000 |
| Batch size | 512 |
| Optimiser | Adam Kingma & Ba (2015) |
| Learning rate | 0.0001 |
| Discount $\gamma$ | 0.98 (0.99 for maze) |
| Activations (unless otherwise stated) | ReLU |
| Target network Polyak smoothing coefficient | 0.01 |

Table 3: Hyperparameters used in all experiments. The settings follow those used in (Jeen et al., 2024).

| Different cost budget | | Cost Budget = 150 | | Cost Budget = 100 | | Cost Budget = 50 | |
|---|---|---|---|---|---|---|---|
| | | Performance | Cost | Performance | Cost | Performance | Cost |
| Walker | | | target $\leq$ 150 | | target $\leq$ 100 | | target $\leq$ 50 |
| | stand | 713±116 | 132±17 | 672±196 | 82±10 | 604±194 | 40±6 |
| | run | 331±149 | 124±19 | 204±154 | 70±24 | 184±153 | 22±15 |
| | walk | 146±12 | 139±12 | 136±64 | 94±4 | 131±34 | 30±16 |
| | Flip | 325±91 | 123±15 | 317±105 | 75±23 | 204±130 | 37±13 |
| Quadruped | | | | | | | |
| | Stand | 734±320 | 135±25 | 737±419 | 89±7 | 400±447 | 36±12 |
| | Roll | 762±216 | 114±24 | 761±153 | 81±16 | 177±69 | 17±13 |
| | Roll Fast | 447±116 | 119±20 | 427±255 | 65±33 | 353±252 | 30±21 |
| | Jump | 624±259 | 122±10 | 542±329 | 91±8 | 179±103 | 11±16 |
| | Escape | 35±32 | 107±36 | 41±33 | 73±25 | 25±17 | 33±17 |
| Maze | | | | | | | |
| | Top Left | 900±450 | 138±6 | 710±463 | 80±14 | 411±379 | 35±9 |
| | Top Right | 128±256 | 140±8 | 125±251 | 66±8 | 101±203 | 32±21 |
| | Bottom Left | 0±0 | 126±21 | 0±0 | 62±22 | 0±0 | 34±23 |
| | Bottom Right | 0±0 | 124±13 | 0±0 | 86±7 | 0±0 | 45±2 |

Table 4: Performance and cost comparison across 13 ExORL tasks under various budget requirements. The objective is not to minimize cost arbitrarily, but rather to utilize it as effectively as possible, ideally approaching the budget without exceeding it. **Our method is capable of satisfying varying budget requirements.**

### A.4.2 STATSTICAL GUARANTEE VIA HOEFFDING BOUND

In Algorithm 3, we set $\epsilon_{\text{latent}} = 0$, which means that the full online sample budget is used to search for the best surrogate threshold $\eta'$ and Lagrange multiplier $\lambda$. Alternatively, $\epsilon_{\text{latent}}$ can be determined using the Hoeffding inequality (van de Geer, 2002; Liu et al., 2025), which guarantees that the constraint will be satisfied with probability at least $1 - \delta$ if

$$\epsilon \leftarrow C_{\max} \sqrt{\frac{\log(2/\delta)}{2n}}.$$

This statistically grounded variant of the algorithm is detailed in Algorithm 3. It aims to provide formal guarantees by setting the threshold using $C_{\max}$, which results in a relatively large $\epsilon$. As a consequence, even policies with costs significantly below the budget can satisfy the constraint, since lower costs make the statistical guarantee easier to achieve. This leads to a more conservative solution that may not fully utilize the available cost budget. Nevertheless, this approach is particularly well-suited for risk-averse users who prioritize reliability over aggressive cost utilization. The corresponding results are shown in Table 5.

| Environment | Ours W/O Hoeffding | | Ours + Hoeffding δ= 0.05 | |
|---|---|---|---|---|
| Walker | Performance | Cost | Performance | Cost |
| stand | 672±196 | 82±10 | 581±219 | 36±43 |
| run | 204±154 | 70±24 | 207±211 | 41±43 |
| walk | 136±64 | 94±4 | 126±77 | 53±38 |
| Flip | 317±105 | 75±23 | 186±110 | 37±48 |
| Quadruped | | | | |
| Stand | 737±419 | 89±7 | 18±5 | 0±0 |
| Roll | 761±153 | 81±16 | 78±140 | 13±27 |
| Roll Fast | 427±255 | 65±33 | 38±48 | 21±41 |
| Jump | 542±329 | 91±8 | 18±15 | 0±0 |
| Escape | 41±33 | 73±25 | 3±3 | 16±26 |
| Maze | | | | |
| Top Left | 710±463 | 80±14 | 703±269 | 66±26 |
| Top Right | 125±251 | 66±8 | 125±250 | 12±27 |
| Bottom Left | 0±0 | 62±22 | 0±0 | 20±45 |
| Bottom Right | 0±0 | 86±7 | 0±0 | 39±45 |

Table 5: Performance and cost comparison across 13 ExORL tasks using the Hoeffding bound with $\delta = 0.05$ in Algorithm 3. **Since the Hoeffding bound provides a statistical guarantee, the results are more conservative compared to those in Table 2.**

