# OpenReview forum: "Unconstrained Models as Constrained Problem Solvers: Duality-Driven Adaptation without Retraining"
_ICLR.cc/2026/Conference — ICLR 2026 Conference Withdrawn Submission_

### Official Review · Reviewer_bDSo · 2025-10-29

**Soundness:** 2
**Presentation:** 2
**Contribution:** 1
**Rating:** 4
**Confidence:** 3

**Summary:**

This paper extends the forward-backward (FB) representation framework to enable zero-shot constrained reinforcement learning. The key contribution is reformulating constrained RL as a latent-space search problem where only the Lagrange multiplier needs to be identified, rather than jointly optimizing both policy and multiplier. The method leverages pre-trained FB models that were unaware of constraints during training and adds an online calibration procedure to handle discrepancies between model estimates and actual costs.

**Strengths:**

- The problem studied in this paper, i.e., zero--shot constrained RL, is of high practical significance.
- The paper proposes solutions to real-world challenges including discount factor mismatches and approximation errors.

**Weaknesses:**

1. The core contribution reduces to using $z_{\lambda}=z_r-\lambda z_c$ with existing FB models plus standard bisection search. While the insight is valuable, the technical contribution is somewhat incremental.
2. Theorem 1 seems like a straightforward corollary of Theorem 2 from (Touati & Ollivier, 2021).
3. It would offer more insight if the comparative evaluation contains baselines that use standard constrained RL agents from scratch.
4. The "zero-shot" claim seems like an exaggeration: the online calibration  process requires 30K samples. This is non-trivial compared to the number of samples used in pre-training (2M).

**Questions:**

1. Line 183: $d$ is not defined before this first usage.
2. Can the number of samples needed by online calibration be reduced?

---

> ### Author Response · Authors · 2025-12-03
> **Rebuttal by Authors**
>
> We thank the reviewer for the feedback. Below, we address the concerns.
>
> ### Weaknesses.1 The core contribution reduces to using  $z_λ=z_r−λz_c$ with existing FB models plus standard bisection search. While the insight is valuable, the technical contribution is somewhat incremental.
> We would like to clarify that although neither bisection nor the FB framework is new on its own, existing approaches rely on retraining a new constrained model, and no prior work demonstrates how a pretrained unconstrained FB model can be used to solve a constrained RL problem without any retraining. This is the core issue our work addresses. While recent studies focus on building increasingly powerful zero-shot (unconstrained) FB models, real-world deployment must accommodate different constraint levels across scenarios, and retraining a constrained model from scratch for each new specification is both computationally expensive and highly unstable, as discussed in the main paper. Our contribution is to show that a pretrained unconstrained FB model can be directly adapted to new cost constraints without retraining, requiring only a small amount of online interaction to calibrate the true (undiscounted) cost. This substantially reduces the practical cost of deploying constrained policies and avoids the failure modes commonly observed in full constrained training.
>
>
> ### Weaknesses.2 Theorem 1 seems like a straightforward corollary of Theorem 2 from (Touati & Ollivier, 2021).
> Theorem 1 is indeed inspired by Theorem 2 of Touati & Ollivier (2021), but it is not a direct restatement. Theorem 2 in Touati & Ollivier establishes that the FB representation can encode optimal policies for unconstrained RL across arbitrary reward functions. In contrast, Theorem 1 extends this expressivity guarantee to the constrained RL setting, where the solution is characterized by a saddle-point of a Lagrangian and depends jointly on both the reward embedding $z_r$ and the cost embedding $z_c$ through the multiplier $λ(z_r,z_c,η)$.
> This extension is nontrivial for three reasons:
> Constrained RL requires representing the correct dual-optimal trade-off.
>  Unlike the unconstrained case, where the optimal policy is indexed solely by a reward vector $z_r​$, the optimal constrained solution lies along a path of modified rewards $z_λ=z_r−λz_c$​. Theorem 1 shows that an FB representation can encode all optimal constrained policies arising from this family of reparameterized rewards, which is not covered by prior results.
> The theorem justifies our key methodological step.
>  The theoretical validation that constrained optimal policies correspond to FB policies indexed by $z_λ$​ is what enables our entire approach, reducing constrained RL to a 1-D search over λ. Without Theorem 1, this reduction would not be theoretically grounded.
> No prior work shows that FB can represent constrained optimal solutions.
>  Touati & Ollivier only guarantee expressivity for unconstrained optimal policies. Theorem 1 fills this gap by proving that FB representations are expressive enough to cover the entire family of constrained-optimal solutions induced by Lagrangian relaxation.
>
> For these reasons, Theorem 1 is not simply a corollary but provides the theoretical foundation for adapting an unconstrained zero-shot FB model to constrained problems without retraining, a capability not addressed in previous work.

---

> > ### Author Response · Authors · 2025-12-03
> > **Rebuttal by Authors**
> >
> > ### Weaknesses.3 It would offer more insight if the comparative evaluation contains baselines that use standard constrained RL agents from scratch.
> > We agree that such baselines are important for contextualizing our contribution, and we have now included results for PPO-Lag [ Ray et al. 2019 ] , TRPO-Lag [ Ray et al. 2019 ] published by OpenAI, and RCPO [ Tessler et al. 2019 ] across all tasks (see Table below), trained with the same number of training steps as our method, since they are standard strong safe RL baseline.
> >
> > - 1. Why standard safe RL baselines were not included initially
> >
> > Standard safe RL baselines (PPO-Lag, TRPO-Lag, RCPO) were not included initially because they require millions of online samples and full retraining, whereas our method operates in a zero-shot/offline + few-shot adaptation regime, making their data requirements fundamentally incompatible.
> >
> > - 2. Full experiments for PPO-Lag, TRPO-Lag, and RCPO
> >
> > Table.2 Safe RL Baselines vs. Ours
> >
> > | Env       | Task         | PPO-Lag Reward | PPO-Lag Cost | TRPO-Lag Reward | TRPO-Lag Cost | RCPO Reward  | RCPO Cost  | Ours Reward | Ours Cost |
> > | --------- | ------------ | -------------- | ------------ | --------------- | ------------- | ------------ | ---------- | ----------- | --------- |
> > | Walker    | Stand        | 158.35±7.56    | 35.1±3.23    | 185.17±8.28     | 26.5±1.23     | 202.91±12.36 | 19.1±2.04  | 672±196     | 82±10     |
> > | Walker    | Walk         | 75.34±4.24     | 30.6±5.21    | 77.35±2.04      | 34.29±5.41    | 76.61±4.22   | 62.88±1.76 | 204±154     | 70±24     |
> > | Walker    | Run          | 56.77±2.87     | 27.51±3.08   | 29.87±5.24      | 47.79±2.09    | 22.63±6.21   | 20.43±4.65 | 136±64      | 94±4      |
> > | Walker    | Flip         | 28.15±2.06     | 1.38±0.011   | 72.58±3.33      | 26.19±2.24    | 80.37±5.47   | 32.8±3.67  | 317±105     | 75±23     |
> > | Quadruped | Stand        | 230.17±40.82   | 23.15±4.00   | 270.56±54.69    | 47.92±15.24   | 285.16±58.77 | 51.36±7.28 | 737±419     | 89±7      |
> > | Quadruped | Roll         | 165.22±36.85   | 17.15±1.84   | 170.46±44.54    | 20.15±1.21    | 173.37±24.99 | 19.82±2.03 | 761±153     | 81±16     |
> > | Quadruped | Roll Fast    | 94.27±7.79     | 9.96±3.27    | 96.5±4.35       | 15.13±4.24    | 99.58±3.99   | 22.65±1.28 | 427±255     | 65±33     |
> > | Quadruped | Jump         | 6.16±0.74      | 0.08±1.06    | 125.17±4.98     | 23.15±2.26    | 169.47±15.68 | 37.63±8.86 | 542±329     | 91±8      |
> > | Quadruped | Escape       | 2.56±0.92      | 0.58±1.27    | 16.77±3.84      | 2.94±0.61     | 4.45±0.19    | 0.28±0.54  | 41±33       | 73±25     |
> > | Maze      | Top Left     | 13.27±2.1      | 73.54±5.82   | 19.36±4.69      | 40.53±6.88    | 24.28±1.23   | 55.05±2.31 | 710±463     | 80±14     |
> > | Maze      | Top Right    | 0.11±0.36      | 19.78±1.63   | 0±0             | 20.68±2.77    | 0±0          | 21.63±2.69 | 125±251     | 66±8      |
> > | Maze      | Bottom Left  | 0±0            | 0.82±0.06    | 0±0             | 0±0           | 0.49±0.3     | 1.32±0.15  | 0±0         | 62±22     |
> > | Maze      | Bottom Right | 0±0            | 4.39±1.88    | 0±0             | 1.63±0.02     | 0±0          | 2.89±0.11  | 0±0         | 86±7      |
> >
> >
> > In conclusion, we observed that safe RL baselines were either over-constrain (cost ≈ 0, underutilizing budget), or under-constrain (cost 40–70+) despite Lagrangian penalties. Either fail to satisfy constraints or become overly conservative. The result confirm that our latent-space approach solves a fundamentally different problem setting (zero-shot adaptation without retraining).
> >
> >
> > [ Tessler et al. 2019 ] Tessler, C., Mankowitz, D.J. and Mannor, S. "Reward Constrained Policy Optimization." International Conference on Learning Representations (2019).
> >
> > [ Ray et al. 2019 ] Ray, Alex, Joshua Achiam, and Dario Amodei. "Benchmarking safe exploration in deep reinforcement learning." arXiv preprint arXiv:1910.01708 7.1 (2019): 2.
> >
> > ### Weaknesses.4 The "zero-shot" claim seems like an exaggeration: the online calibration process requires 30K samples. This is non-trivial compared to the number of samples used in pre-training (2M).
> > We acknowledge the potential for confusion created by our use of “zero-shot” in several places. Let us clarify what we aim to achieve in this work. As highlighted in the title, the goal is to solve constrained RL problems **without retraining**; that is, the weights of the main model (the task-conditioned predictor $F$, the reward-agnostic embedding $B$, and the policy network) remain frozen. However, the model can accept task-specific inputs, denoted $z_\lambda$, which are selected using a small interaction budget. In practice, we find that most tasks converge within around 10k online samples, and only a few rare cases require up to 30k. We believe this represents a reasonable and practical cost for deploying an existing model under new safety or budget requirements, and does not contradict our intended “no-retraining” setting.

---

> > > ### Author Response · Authors · 2025-12-03
> > > **Rebuttal by Authors**
> > >
> > > ### Questions.1 Line 183: d  is not defined before this first usage.
> > > The symbol 𝑑 refers to the dimension of the latent space used in the FB representations. We have revised the main paper to explicitly define 𝑑 at its first occurrence, and the clarification is highlighted in blue in the updated manuscript.
> > >
> > > ### Questions.2 Can the number of samples needed by online calibration be reduced?
> > >  In practice, we find that most tasks converge within around 10k online samples, and only a few rare cases require up to 30k. We believe this represents a reasonable and practical cost for deploying an existing model under new safety or budget requirements, and does not contradict our intended “no-retraining” setting.

---

### Official Review · Reviewer_1nsU · 2025-10-30

**Soundness:** 3
**Presentation:** 2
**Contribution:** 3
**Rating:** 4
**Confidence:** 4

**Summary:**

The paper aims to achieve zero-shot adaptation for different constraint configurations by using a forward-backward (FB) framework that models the optimal policies for a set of potential rewards. Specifically, it searches for an appropriate Lagrangian parameter based on value estimation, and this parameter, together with the reward latent and constraint latent, forms an index to constraint-satisfying optimal policies. In addition, the paper proposes using a few online samples to finetune the Lagrangian parameter to mitigate the gap between the real cost return and the cost value estimate. Experiments on ExORL benchmark datasets demonstrate the method’s ability to achieve constraint-satisfying policies through the FB framework and online calibration.

**Strengths:**

The motivation of using the FB framework to achieve zero-shot constraint adaptation is relatively novel and intuitively reasonable. Specifically, the potential of learning safe behaviors for various constraints even without explicit constraint functions during training is impressive.

**Weaknesses:**

- Descriptions and formulations should be further improved.
    - I recommend substantial revisions to Section 3, as many notations of FB are not clearly introduced. For example, the definitions of $ds’$ and $ρ$, the connection of $X$ in Eq.(4) with other formulations, and the operational details of $Q_r^* = Mr$, $Q_r^*(s,a) ≈ (F(s,a,·))B(·)^T r$, and $M ≈ FB$.
    - In addition, the implementation details for training $F$ and $B$ should be provided, or at least linked to the appendix for reference.
    - The $\overline F$ in Eq.(15) appears to differ from the one in Section 4.1, so this notation should be reclarified.
- The choice of baselines is limited and lacks comparisons with standard safe RL baselines. Several existing works [1–3] have addressed safe RL under various constraint thresholds, and these should be discussed and compared.

-  Although the paper claims zero-shot adaptation for both new cost functions and thresholds, the experiments only consider varying thresholds.

- The assumption of access to 3k online samples and their cost returns for fine-tuning is rather strong. While it intuitively improves performance for any algorithm that models diverse policies, it is demanding in practice and somewhat contradicts the zero-shot setting claimed in the paper.

[1] Liu, Zuxin, et al. "Constrained decision transformer for offline safe reinforcement learning." *International conference on machine learning*. PMLR, 2023.

[2] Lin, Qian, et al. "Safe offline reinforcement learning with real-time budget constraints." *International Conference on Machine Learning*. PMLR, 2023.

[3] Yao, Yihang, et al. "Constraint-conditioned policy optimization for versatile safe reinforcement learning." *Advances in Neural Information Processing Systems* 36 (2023): 12555-12568.

**Questions:**

- There are some environments specifically designed for constraint RL, such as Safety-Gym [1] and OSRL [2]. I believe they are also suitable for evaluating different thresholds or even different constraint functions. Could you clarify why these benchmarks were not used?
- I am a bit confused about the CRL_FB results in the table. While I understand that cost value estimation errors can lead to instability and constraint violations, why does the method exhibit extremely conservative behaviors? Could you elaborate on this point?
- I recommend reporting a plot of the relationship between $\lambda$ and $Q_c$, which could help verify their correlation.


[1] Ji, Jiaming, et al. "Safety gymnasium: A unified safe reinforcement learning benchmark." *Advances in Neural Information Processing Systems* 36 (2023): 18964-18993.

[2] Liu, Zuxin, et al. "Datasets and benchmarks for offline safe reinforcement learning." *arXiv preprint arXiv:2306.09303* (2023).

---

> ### Author Response · Authors · 2025-12-03
> **Rebuttal by Authors**
>
> We appreciate the reviewer’s thoughtful comments and for identifying several weaknesses and questions. Our detailed responses to each point are outlined individually below:
>
> ### Weaknesses.1 Descriptions and formulations should be further improved.
> We have added a dedicated supplemental section that provides a complete and self-contained explanation of the clarified symbols in Sections 3 and 4.1, along with additional training details; see Appendix for details. All amendments are highlighted in blue for easy reference.
>
> ### Weaknesses.2 The choice of baselines is limited and lacks comparisons with standard safe RL baselines.
> We agree that such baselines are important for contextualizing our contribution, and we have now included results for PPO-Lag [ Ray et al. 2019 ] , TRPO-Lag [ Ray et al. 2019 ] published by OpenAI, and RCPO [ Tessler et al. 2019 ] across all tasks (see Table below), trained with the same number of training steps as our method, since they are standard strong safe RL baseline.
>
> - 1. Why standard safe RL baselines were not included initially
>
> Standard safe RL baselines (PPO-Lag, TRPO-Lag, RCPO) were not included initially because they require millions of online samples and full retraining, whereas our method operates in a zero-shot/offline + few-shot adaptation regime, making their data requirements fundamentally incompatible.
>
> - 2. Full experiments for PPO-Lag, TRPO-Lag, and RCPO
>
> Table.2 Safe RL Baselines vs. Ours
>
> | Env       | Task         | PPO-Lag Reward | PPO-Lag Cost | TRPO-Lag Reward | TRPO-Lag Cost | RCPO Reward  | RCPO Cost  | Ours Reward | Ours Cost |
> | --------- | ------------ | -------------- | ------------ | --------------- | ------------- | ------------ | ---------- | ----------- | --------- |
> | Walker    | Stand        | 158.35±7.56    | 35.1±3.23    | 185.17±8.28     | 26.5±1.23     | 202.91±12.36 | 19.1±2.04  | 672±196     | 82±10     |
> | Walker    | Walk         | 75.34±4.24     | 30.6±5.21    | 77.35±2.04      | 34.29±5.41    | 76.61±4.22   | 62.88±1.76 | 204±154     | 70±24     |
> | Walker    | Run          | 56.77±2.87     | 27.51±3.08   | 29.87±5.24      | 47.79±2.09    | 22.63±6.21   | 20.43±4.65 | 136±64      | 94±4      |
> | Walker    | Flip         | 28.15±2.06     | 1.38±0.011   | 72.58±3.33      | 26.19±2.24    | 80.37±5.47   | 32.8±3.67  | 317±105     | 75±23     |
> | Quadruped | Stand        | 230.17±40.82   | 23.15±4.00   | 270.56±54.69    | 47.92±15.24   | 285.16±58.77 | 51.36±7.28 | 737±419     | 89±7      |
> | Quadruped | Roll         | 165.22±36.85   | 17.15±1.84   | 170.46±44.54    | 20.15±1.21    | 173.37±24.99 | 19.82±2.03 | 761±153     | 81±16     |
> | Quadruped | Roll Fast    | 94.27±7.79     | 9.96±3.27    | 96.5±4.35       | 15.13±4.24    | 99.58±3.99   | 22.65±1.28 | 427±255     | 65±33     |
> | Quadruped | Jump         | 6.16±0.74      | 0.08±1.06    | 125.17±4.98     | 23.15±2.26    | 169.47±15.68 | 37.63±8.86 | 542±329     | 91±8      |
> | Quadruped | Escape       | 2.56±0.92      | 0.58±1.27    | 16.77±3.84      | 2.94±0.61     | 4.45±0.19    | 0.28±0.54  | 41±33       | 73±25     |
> | Maze      | Top Left     | 13.27±2.1      | 73.54±5.82   | 19.36±4.69      | 40.53±6.88    | 24.28±1.23   | 55.05±2.31 | 710±463     | 80±14     |
> | Maze      | Top Right    | 0.11±0.36      | 19.78±1.63   | 0±0             | 20.68±2.77    | 0±0          | 21.63±2.69 | 125±251     | 66±8      |
> | Maze      | Bottom Left  | 0±0            | 0.82±0.06    | 0±0             | 0±0           | 0.49±0.3     | 1.32±0.15  | 0±0         | 62±22     |
> | Maze      | Bottom Right | 0±0            | 4.39±1.88    | 0±0             | 1.63±0.02     | 0±0          | 2.89±0.11  | 0±0         | 86±7      |
>
>
> In conclusion, we observed that safe RL baselines were either over-constrain (cost ≈ 0, underutilizing budget), or under-constrain (cost 40–70+) despite Lagrangian penalties. Either fail to satisfy constraints or become overly conservative. The result confirm that our latent-space approach solves a fundamentally different problem setting (zero-shot adaptation without retraining).
>
>
> [ Tessler et al. 2019 ] Tessler, C., Mankowitz, D.J. and Mannor, S. "Reward Constrained Policy Optimization." International Conference on Learning Representations (2019).
>
> [ Ray et al. 2019 ] Ray, Alex, Joshua Achiam, and Dario Amodei. "Benchmarking safe exploration in deep reinforcement learning." arXiv preprint arXiv:1910.01708 7.1 (2019): 2.

---

> > ### Author Response · Authors · 2025-12-03
> > **Rebuttal by Authors**
> >
> > ### Weaknesses.3 Although the paper claims zero-shot adaptation for both new cost functions and thresholds, the experiments only consider varying thresholds.
> >
> > To verify this experimentally, we added a new ablation in which we change the cost function itself, not only the constraint threshold. We constructed a family of five distinct cost functions and named them C1~C5, corresponding to different velocity- or joint-related safety features in Walker. All conditions use ETA = 100, but the underlying cost mapping is different in every column.
> >
> > Table 3. Different Cost Constraints (ETA = 100)
> >
> > | Env    | Task  | C1 Reward | C1 Cost | C2 Reward | C2 Cost | C3 Reward | C3 Cost | C4 Reward | C4 Cost | C5 Reward | C5 Cost |
> > | ------ | ----- | --------------- | ------------- | --------------- | ------------- | --------------- | ------------- | --------------- | ------------- | --------------- | ------------- |
> > | walker | stand | 795.57          | 97.54         | 792.38          | 90.89         | 792.38          | 99.43         | 790.54          | 91.19         | 848.35          | 72.15         |
> > | walker | walk  | 283.92          | 93.93         | 599.05          | 98.37         | 179.07          | 91.39         | 634.51          | 80.01         | 588.02          | 92.08         |
> > | walker | run   | 34.69           | 75.24         | 271.26          | 98.06         | 140.99          | 80.18         | 116.85          | 94.25         | 287.59          | 57.90         |
> > | walker | flip  | 214.35          | 52.32         | 409.60          | 91.24         | 518.25          | 83.88         | 46.80           | 57.25         | 371.08          | 99.59         |
> >
> > As result above, our method successfully adapts to completely different cost functions, each column corresponds to a new definition of cost that differs in semantics and magnitudes. The results confirm that our approach maintains constraint satisfaction and effective performance across multiple, semantically different cost signals without retraining, thereby validating our claim.
> >
> > ### Weaknesses.4 The assumption of access to 3k online samples and their cost returns for fine-tuning is rather strong. While it intuitively improves performance for any algorithm that models diverse policies, it is demanding in practice and somewhat contradicts the zero-shot setting claimed in the paper.
> >
> > We acknowledge the potential for confusion created by our use of “zero-shot” in several places. Let us clarify what we aim to achieve in this work. As highlighted in the title, the goal is to solve constrained RL problems **without retraining**; that is, the weights of the main model (the task-conditioned predictor $F$, the reward-agnostic embedding $B$, and the policy network) remain frozen. However, the model can accept task-specific inputs, denoted $z_\lambda$, which are selected using a small interaction budget. In practice, we find that most tasks converge within around 10k online samples, and only a few rare cases require up to 30k. We believe this represents a reasonable and practical cost for deploying an existing model under new safety or budget requirements, and does not contradict our intended “no-retraining” setting.

---

> ### Author Response · Authors · 2025-12-03
> **Rebuttal by Authors**
>
> ### Questions.1 I am a bit confused about the CRL_FB results in the table. While I understand that cost value estimation errors can lead to instability and constraint violations, why does the method exhibit extremely conservative behaviors? Could you elaborate on this point?
>
> Constrained RL is known to be substantially more difficult than standard RL, prior works report that even solving a single constrained MDP often requires 6–10× more data (e.g., 6M–10M samples for MuJoCo tasks). This difficulty arises because the model must simultaneously optimize the reward while ensuring that the cost value function is accurate enough to enforce the constraint, which is highly sensitive to estimation errors.
> In the case of CRL_FB, the challenge is amplified: instead of learning to satisfy a single cost function, the model must learn representations that remain valid across many different latent cost vectors $z_c$​. To guarantee feasibility for all possible $z_c$​, the learned policy tends to shift toward behaviors that are safe under every cost configuration. This leads to overly conservative behavior, where the model avoids almost all potentially costly actions, even when those actions are necessary for achieving meaningful task performance.
> This behavior is consistent with what is well-documented in primal–dual constrained RL: when faced with uncertainty in cost estimation or multiple possible constraint directions, the dual variable λ\lambdaλ grows aggressively, pushing the policy toward trivial “do-nothing” solutions that avoid all costs. Our empirical results reflect this instability: CRL_FB satisfies the constraint but collapses into extremely conservative, low-return policies.
> In contrast, our method avoids this issue because we do not train a multi-constraint model. Instead, we reuse the pretrained FB model and search for the appropriate λ\lambdaλ only for the single cost configuration relevant at deployment, which eliminates the need for the policy to be feasible across many unseen constraints.
>
> ### Questions.2 I recommend reporting a plot of the relationship between $\lambda$ and $Q_c$, which could help verify their correlation.
>
> In response, we have conducted an empirical sweep over λ during the Lagrange multiplier search and recorded the corresponding latent-space cost estimate $Q_c(\lambda)$. The data below are from the Walker–Walk task:
>
> 1.Overview of calibration step:
>
> During calibration, the key step is updating $\lambda$ according to the predicted latent cost, since the pretrained FB model is unconstrained and does not specify which $\lambda$ satisfies a given budget. We iteratively adjust $\lambda$ by checking whether $Q_c(\lambda)$ is above or below the surrogate threshold:
>
> - $Q_c(\lambda)$ too high → increase $\lambda$
>
> - $Q_c(\lambda)$ too low → decrease $\lambda$
>
> Because $Q_c(\lambda)$ decreases monotonically with $\lambda$, this process behaves like a bisection search in latent space. After locating a $\lambda$ whose latent cost matches the target, we perform a small number of rollouts to align latent and real costs. Thus, calibration does not retrain the policy—it simply selects the $\lambda$ that yields a constraint-satisfying policy.
>
> 2. The record of $\lambda$ with $Q_c$ over iteration:
>
> Table 4. λ–Qc over iterations record
>
> | Iteration |        λ |      Q_c |
> | --------: | -------: | -------: |
> |         1 | 2.763907 | -922.195 |
> |         2 | 1.381953 | -392.826 |
> |         3 | 0.690977 | -26.9214 |
> |         4 | 1.036465 | -207.914 |
> |         5 | 1.209209 | -313.589 |
> |         6 | 1.122837 | -264.379 |
> |         7 | 1.079651 | -235.689 |
>
> These points as above shows a smooth, unimodal downward curve.

---

### Official Review · Reviewer_4fa7 · 2025-11-02

**Soundness:** 2
**Presentation:** 2
**Contribution:** 2
**Rating:** 2
**Confidence:** 3

**Summary:**

The paper extends forward–backward (FB) representations to constrained reinforcement learning by embedding reward and cost into a shared latent space and using a Lagrangian-duality reparameterization to enable zero-shot, constraint-aware policy inference without retraining. Specifically, author apply a bisection method to identify the appropriate  Lagrange multiplier. I think the proposed method is novel and have several questions about the experimental setup and specific implementation details, which I outline below.

**Strengths:**

1. This paper convert the Lagrangian dual optimization problem into a direct search over λ, which is simple and effective.
2. The paper is clearly structured, and I was able to grasp its content quickly.

**Weaknesses:**

1. The paper uses a bisection search over $\lambda$, but it does not prove that this procedure yields the optimal $\lambda$.  The bisection routine implicitly assumes that the surrogate cost  $\hat{C}_{\text{latent}}(\lambda) = \overline{F}(\cdot, z_\lambda)^\top z_c$  is monotone in $\lambda$, but the paper does not establish such monotonicity; policy switches can make  $\hat{C}_{\text{latent}}(\lambda)$ non-monotonic.
2. The **Online Calibration** stage requires extra online sampling, whereas the baselines are evaluated without such an interaction pass. It makes the comparison not strictly fair. The author didn’t mention this point in experiments.

3. The experiments do not include a comparison against directly applying the original Lagrangian dual optimization. Consequently, Limitation 1 is not empirically substantiated.

**Questions:**

1. The proposed method is based on FB—which inherently relies on discounted returns—and thus does not fundamentally resolve Limitation 2.
2. The algorithm’s performance without online calibration has not been evaluated.
3. The $\lambda$-search procedure lacks theoretical justification. The parameterization $z_{\lambda} = z_r - \lambda z_c$  constrains the policy space —and is this form essentially optimal?

---

> ### Author Response · Authors · 2025-12-03
> **Rebuttal by Authors**
>
> We thank the reviewer for the feedback. Below, we address the concerns.
>
> ### “Monotonicity and bisection routine”
>
> As already noticed by the reviewers, bisection search is one of the classical techniques for Lagrangian-based constrained optimization ([ Kumar et al. 2021 ] and [ Ruan et al. 2020 ]) with long history. Hence, it's a natural first attempt to pull it out in our optimization problem. Now, it’s true that with approximation in the latent surrogate, we most likely face the difficulties of a non-monotonic objective. Nevertheless, our current procedure, even though it can be suboptimal, already consistently produces budget-satisfying policies with strong performance; strengthening the selection step can only preserve feasibility and, if anything, improve returns.
>
> [ Kumar et al. 2021 ] Kumar, Tej, and Krishnan Suresh. "Direct lagrange multiplier updates in topology optimization revisited." Structural and Multidisciplinary Optimization 63.3 (2021): 1563-1578.
> [ Ruan et al. 2020 ] Ruan, Guangchun, et al. "Neural-network-based Lagrange multiplier selection for distributed demand response in smart grid." Applied Energy 264 (2020): 114636.
>
>
> ### “Comparison against directly applying the original Lagrangian dual optimization & Limitation 1”
>
> We would like to clarify that CRL_FB is the primal-dual Lagrangian optimization baseline, implemented following the standard constrained RL formulation described in Section 4.1 of the paper. That is, CRL_FB performs joint updates over the policy $\pi$ and the Lagrange multiplier $\lambda$ using alternating gradient steps, exactly matching the original Lagrangian dual optimization procedure.
>
> Thus, the experimental results do substantiate Limitation 1: the shortcomings of direct Lagrangian dual optimization manifest in CRL_FB, validating the need for our more stable latent-space search procedure.
>
> ### “FB-based method and limitation 2”
>
> We agree that FB’s internal estimates are discounted. This is precisely why we introduce the online calibration step: during calibration, the constraint is evaluated using the true, undiscounted cumulative cost collected directly from the environment, rather than relying on the discounted surrogate cost estimated by the latent model. In effect, the discounted FB estimates are used only to propose a candidate $\lambda$, but the final constraint satisfaction is accepted only if it passes an empirical budget check based on undiscounted costs (with a small number of rollouts). Therefore, while the FB representation itself inherits Limitation 2, our calibration mechanism removes the reliance on discounted estimates at evaluation time, meaning this limitation is no longer fundamental to the proposed approach.
>
> ### “The algorithm’s performance without online calibration has not been evaluated.”
>
> We would like to point out that the FB baseline (methods with _FB in their names) corresponds to our method without the online calibration step. Thus, the experiments already include and report performance of the no-calibration setting through the FB baseline, and they clearly show why calibration is required for reliable constraint satisfaction.
>
> On the other hand, we could also apply the same calibration procedure to the CRL_FB baseline. The results are shown in Table 1 (CRL_FB + calibration).
>
> Table.1 CRL_FB+calibration
>
> | Env    | Task  | Reward | Cost |
> | ------ | ----- | ------ | ---- |
> | walker | stand | 132.65 | 0    |
> | walker | walk  | 22.9   | 0    |
> | walker | run   | 22.19  | 0    |
> | walker | flip  | 23.14  | 0    |
>
> We demonstrate that when CRL_FB is given the exact same calibration advantage as our method, it ends up converging to an overly conservative policy, driving the cost to zero across all tasks, thereby failing to utilize the available budget ($\eta = 100$). Moreover, due to such extreme conservatism, CRL_FB achieves very low reward (typically less than 20% of our method).
>
> These results confirm that calibration does not fundamentally alter the performance gap:
> our approach remains substantially more effective at balancing task performance and constraint satisfaction.

---

### Official Review · Reviewer_LsEu · 2025-11-06

**Soundness:** 2
**Presentation:** 3
**Contribution:** 2
**Rating:** 4
**Confidence:** 3

**Summary:**

The paper proposes a post-hoc way to impose budget (cost) constraints on a pre-trained forward–backward (FB) model at deployment time. It linearly combines reward and cost embeddings with a scalar $\lambda$ in latent space, searches $\lambda$ (e.g., via bisection) to meet a target budget, and uses a light online calibration step to correct the gap between latent and realized costs. Empirically, the method is evaluated on continuous-control tasks and compared against FB variants and a primal–dual baseline, showing a favorable performance–compliance trade-off.

**Strengths:**

Simple & practical: Turns constrained control into a 1-D search over  $\lambda$  without retraining; easy to bolt onto existing FB models.

Low deployment overhead: Only post-hoc inference + brief online calibration; attractive for real systems where retraining is costly.

Empirical coverage: Multiple tasks, budgets, and ablations (including a Hoeffding-style variant) help illustrate behavior across regimes.

Clear motivation: Addresses discount/representation mismatch between latent and realized costs and proposes a concrete calibration mechanism.

**Weaknesses:**

* **“Zero-shot” vs. online calibration:** The approach depends on non-trivial online interactions for calibration and ($\lambda$) selection. This weakens the “zero-shot” claim; please quantify how much those interactions matter (strict zero-shot vs. few-shot vs. your full setting).
* **Monotonicity/robustness of the ($\lambda$  $\mapsto$) cost curve:** The method assumes near-monotonic behavior to justify bisection, but function approximation noise can break this. The paper would benefit from conditions, diagnostics, and fallbacks for non-monotone cases.
* **Theory depth:** Current analysis mostly repackages Lagrangian ideas into latent space; tighter guarantees under approximation error and finite data would strengthen novelty.

**Questions:**

# Questions

1. **Positioning vs. BCORLE ((\lambda)).**
   BCORLE uses **(\lambda)-generalization** to reuse policies across different budgets in an offline coupon-allocation setting. Could you clarify where your method aligns or diverges—e.g., training regime (offline vs. pre-trained FB), reliance on online calibration, supported constraint types, and deployment-time adaptability?

2. **Stronger baselines.**
   Baselines feel a bit light—could you consider adding a stronger reference point such as **TREBI (ICML 2023)** for real-time budget constraints?

3. **Monotonicity for bisection.**
   Your approach implicitly assumes the realized cost is (approximately) monotone in ($\lambda$ ). Could you (i) state conditions under which this holds with function approximation and finite data; (ii) report how often violations occur, with representative cost–($\lambda$ ) plots and failure cases?

4. **($\lambda$ ) bounds & stopping.**
   How are search bounds and stopping criteria chosen across tasks? Is there a task-agnostic or adaptive rule, and how sensitive are outcomes to these choices?

5. **Generalization.**
   Does a (\lambda) tuned for one task/domain transfer to related tasks, or must calibration be repeated each time?

---

> ### Author Response · Authors · 2025-12-03
> **Rebuttal by Authors**
>
> We thank the reviewer for the feedback. Below, we address the concerns.
>
> ### “Zero-shot” vs. online calibration
> We acknowledge the potential for confusion created by our use of “zero-shot” in several places. Let us clarify what we aim to achieve in this work. As highlighted in the title, the goal is to solve constrained RL problems **without retraining**; that is, the weights of the main model (the task-conditioned predictor $F$, the reward-agnostic embedding $B$, and the policy network) remain frozen. However, the model can accept task-specific inputs, denoted $z_\lambda$, which are selected using a small interaction budget. Hence, the online calibration step performs this hyperparameter selection - not any update of model weights - and therefore falls under a “few-shot” usage of data.
> This calibration is necessary because FB’s internal estimates are **discounted**, whereas constraint satisfaction is judged by **undiscounted** cumulative cost. Accordingly, deployment is gated by an empirical **undiscounted** budget check using a few rollouts. In practice, our selection procedure typically resolves $\lambda$ within $\sim$ 10k environment steps (rare cases up to $\sim$ 30k), versus $\sim$ 2M steps for full retraining. For completeness, the *FB* baselines implement the same pipeline **without** calibration (zero additional interaction) and consistently fail to guarantee budgets, underscoring why calibration is required.
> In the revision, we will (i) clarify the setting up front more explicitly and (ii) emphasize per-task interaction budgets alongside full-retraining costs.
> ### Monotonicity/robustness of the $\lambda$ cost curve
> As already noticed by the reviewers, bisection search is one of the classical techniques for Lagrangian-based constrained optimization ([ Kumar et al. 2021 ] and [ Ruan et al. 2020 ]) with long history. Hence, it's a natural first attempt to pull it out in our optimization problem. Now, it’s true that with approximation in the latent surrogate, we most likely face the difficulties of a non-monotonic objective. Nevertheless, our current procedure, even though it can be suboptimal, already consistently produces budget-satisfying policies with strong performance; strengthening the selection step can only preserve feasibility and, if anything, improve returns.
> [ Kumar et al. 2021 ] Kumar, Tej, and Krishnan Suresh. "Direct lagrange multiplier updates in topology optimization revisited." Structural and Multidisciplinary Optimization 63.3 (2021): 1563-1578.
> [ Ruan et al. 2020 ] Ruan, Guangchun, et al. "Neural-network-based Lagrange multiplier selection for distributed demand response in smart grid." Applied Energy 264 (2020): 114636.
>
> ### $\lambda$ bound and stopping
> The stopping criterion is set to when the process "converge", i.e., does not change a lot, with a hard cap. As  mentioned in the reply to “zero-shot vs online calibration”, the selection procedure typically resolves within $\sim$ 10k environment steps, but a few require slightly bigger.
>
> ### Stronger baseline (TREBI)
> Thank you for the pointer. We agree TREBI is relevant to constrained/safe RL, and we will include the comparison in the paper. Our understanding is that TREBI is designed for per-task offline adaptation: it performs no online interaction, but uses task-specific offline datasets with multiple (successful) trajectories from policies trained for that task (20k for Pendulum and 1M otherwise). By contrast, our method trains a single reward-free FB model once and adapts to new reward specifications and cost budgets using a small amount of online calibration (20k). Despite not being directly comparable, we are also working on adapting TREBI to our setup; the experiment is currently ongoing.
>
> ### Generalization
> In general, $\lambda$ is **task-specific**: it depends on the downstream reward and cost budget. Therefore, to guarantee budget satisfaction, we perform a brief calibration for each new task specification. That being said, $\lambda$ often transfers well as a warm start across closely related settings, and we need only a few additional rollouts to re-confirm feasibility.
> If one does not require budget guarantees, the prior zero-shot FB method can adapt to new tasks without calibration, but it provides no constraint satisfaction guarantee. Our focus is maintaining guarantees with a small, per-task calibration while keeping the model weights frozen (no retraining).

---

### Note · Authors · 2026-01-20

I have read and agree with the venue's withdrawal policy on behalf of myself and my co-authors.